# Precise Modeling of the Protective Effects of Quercetin against Mycotoxin via System Identification with Neural Networks

**DOI:** 10.3390/ijms20071725

**Published:** 2019-04-08

**Authors:** Changju Yang, Entaz Bahar, Shyam Prasad Adhikari, Seo-Jeong Kim, Hyongsuk Kim, Hyonok Yoon

**Affiliations:** 1Division of Electronics Engineering and Research Center for Intelligent Robots, Chonbuk National University, Jeonju 54896, Korea; ychangju@jbnu.ac.kr (C.Y.); all.shyam@gmail.com (S.P.A.); scott3554@naver.com (S.-J.K.); 2College of Pharmacy and Research Institute of Pharmaceutical Sciences, Gyeongsang National University, Jinju 52828, Korea; entaz_bahar@yahoo.com

**Keywords:** cell cytotoxicity, mycotoxins, quercetin, artificial neural networks, computational modeling

## Abstract

Cell cytotoxicity assays, such as cell viability and lactate dehydrogenase (LDH) activity assays, play an important role in toxicological studies of pharmaceutical compounds. However, precise modeling for cytotoxicity studies is essential for successful drug discovery. The aim of our study was to develop a computational modeling that is capable of performing precise prediction, processing, and data representation of cell cytotoxicity. For this, we investigated protective effect of quercetin against various mycotoxins (MTXs), including citrinin (CTN), patulin (PAT), and zearalenol (ZEAR) in four different human cancer cell lines (HeLa, PC-3, Hep G2, and SK-N-MC) in vitro. In addition, the protective effect of quercetin (QCT) against various MTXs was verified via modeling of their nonlinear protective functions using artificial neural networks. The protective model of QCT is built precisely via learning of sparsely measured experimental data by the artificial neural networks (ANNs). The neuromodel revealed that QCT pretreatment at doses of 7.5 to 20 μg/mL significantly attenuated MTX-induced alteration of the cell viability and the LDH activity on HeLa, PC-3, Hep G2, and SK-N-MC cell lines. It has shown that the neuromodel can be used to predict the protective effect of QCT against MTX-induced cytotoxicity for the measurement of percentage (%) of inhibition, cell viability, and LDH activity of MTXs.

## 1. Introduction

Cell cytotoxicity assays have a central role in toxicology studies in the assessment of the in vivo toxic potential of pharmaceutical chemical agents based on in vitro cell cytotoxicity studies [1]. In vitro cell cytotoxicity assays are generally used for drug screening to detect whether the test molecules have effects on cell proliferation or display direct cytotoxic effects. By using this concept, it is possible to measure the protective effect of pharmaceutical lead compounds as well. Two cytotoxicity assays, including cell viability and lactate dehydrogenase (LDH) leakage assays, are widely used in vitro toxicology studies. Cell viability is an important indicator for understanding the underlying mechanisms of certain genes, proteins, and pathways involved cell survival or death after exposing to toxic agents [2]. The LDH leakage assay is based on the measurement of LDH activity in the extracellular medium, where the loss of intracellular LDH and its release into the culture medium is an indicator of irreversible cell death due to cell membrane damage [3]. In vitro cell viability and LDH leakage assays play a crucial role in predictive toxicology in assessing the toxicity of chemicals. However, the quality of in vitro cytotoxicity data fluctuates dramatically due to the variation of experimental modeling, data collection, data analysis, and interpretation. In addition, there has been a remarkable increase in the amount of pharmaceutical compounds that have to assess potential cytotoxic effect for toxicological evaluation [4]. Particularly from the point of view of research laboratories and pharmaceutical industries, the role of precise modeling for cytotoxicity studies is essential for successful drug development because toxicity still remains one of the major factors for failure in the drug discovery [5].

Mycotoxins (MTXs) are toxic developing in many foodstuffs which are responsible for creating a worldwide problem in various agricultural commodities [6,7,8]. Finding out a drug candidate that can reduce or minimize MTX-induced toxicity is a great challenge for researchers because MTXs exhibit high diversified toxic effects in humans and animals [9,10].

Quercetin (QCT) (3,3′,4′,5,7-pentahydroxyflavone) is a natural dietary compound that possess strong anti-oxidant and exhibited beneficial effect against major classes of MTXs in both in vitro and in vivo experiments [11]. It has been reported that QCT inhibits MTX-induced cytotoxicity and oxidative stress in the liver of rats [12]. The effect QCT as well as other many natural compounds against MTXs still remains unknown. In this study, the effect of QCT is investigated with the help of neuro modeling about several MTXs, such as Citrinin (CTN), Patulin (PAT), and Zearalenol (ZEAR).

Artificial neural networks (ANNs) belong to a class of Artificial Intelligence (AI) tools which are inspired by neuroscience and the architecture of human brain. Similar to the biological neural networks, ANNs are capable of parallel processing using large number of neurons and the learning is conducted based on the given data [13,14,15,16,17]. ANNs have been widely used for problems in pattern detection and classification [18,19,20,21,22].

Neuromodeling is a methodology for building models of systems using ANN via learning the relationship between the inputs and their corresponding outputs. This modeling method is especially useful for applications where the relationship between the system input and output is unknown and/or highly nonlinear. The behavior of the original system can be reproduced via an ANN-based model learned using sparsely measured local data of the system. This process is also called ANN-based system identification [23].

The aim of our study is to develop a precise model for the protective effect of QCT against MTX-induced cytotoxicity in four different human cell lines (HeLa, PC-3, Hep G2, and SK-N-MC) using the ANN-based system identification method. Toward this end, two ANN models were designed to predict the protective effect of QCT against MTX-induced cytotoxicity that was evaluated with the measurement of percentage (%) of inhibition, cell viability, and LDH activity of MTXs (CTN, PAT, and ZEAR) in HeLa, PC-3, Hep G2, and SK-N-MC.

## 2. System Identification Using Artificial Neural Network

### 2.1. Neuromodeling via System Identification

System identification is a methodology for building models of a dynamic system using measurements of the system’s input and output signals. It also includes the optimal design of experiments for efficiently generating informative data for fitting such models as well as model reduction. In this study, a neural network approach has been used for system identification. Specifically, multilayer neural networks and back propagation [24] learning has been employed. Figure 1 shows a diagram to explain the concept of system identification.

The system to be modeled is placed in parallel with a neural network with nonlinear learning capability. That input signal **x** is given to both the system and the neural network and the output of the system is taken as the desired response **d** for training the neural network. The objective of system identification is then to build a neural network whose response **o** matches to the response **y** of the system for a given set of inputs **x**. To achieve this objective input signals are presented repeatedly to the network and the norm of the error vector ‖d−o‖, is minimized using back-propagation algorithm, which iteratively adjusts the weights of the neural network until its response is close to the desired system response. 

### 2.2. Multilayer Neural Networks

A multilayer neural network (MNN), as shown in Figure 2, is a class of feed-forward artificial neural network that consists of an input layer, an output layer, and at least one hidden layer in between them [16,17]. Except for the input layer, the basic element in each layer is a node called neuron. The input layer receives inputs **x** from the external world. This data is weighted with different synaptic weights {***w_ji_***} and feed-forwarded to the hidden layer. Each neuron in the hidden layer sums the weighted inputs it receives from its preceding layer and applies a nonlinear transfer function also called “activation function” before passing them as inputs to the output layer. The output layer neurons perform similar computation and yield the output values ***y*** of the neural network. The output of the hidden and output neurons can be expressed as follows.
(1)$$z_{j}=f_{h}(\sum_{i}x_{i}.w_{ji}+b_{j})$$
(2)$$y_{k}=f_{o}(\sum_{j}z_{j}.w_{kj}+b_{k})$$
where ***f_h_*** and ***f_o_*** are the activation functions, and ***b_j_*** and ***b_k_*** are the bias of the hidden and output layer, respectively. In this study the sigmoid activation function is used:(3)$$f(a)=1/(1+e^{-a})$$

The multiple layers and nonlinear activation enable the MNN to learn a mapping of any complexity. The network is learned based on repeated presentations of the training samples and iterative adjustments of the weights using back-propagation algorithm.

### 2.3. System Identification via Learning with Back-propagation

Assume that the error function of the network shown in Figure 2 is defined as
(4)$$E=\sum_{p}{\left(y_{p}-t_{p}\right)}^{2}$$
where ***y*** is the network output and ***t*** is the desired target and the error is computed over all the data points ***p***. Using (1) and (2), (4) can be written as
(5)$$E=\sum_{p}{\left(f_{o}\left(\sum_{j}w_{kj}.f_{h}\left(\ldots (X^{p})\right)\right)-t_{p}\right)}^{2}$$

If all **W** are chosen appropriately for all the patterns, then the error E will approach close to zero. At this situation, the system can produce output values close to the target values for all the inputs. At this state the network is regarded as completely learned and we can say that the function $f_{o}\left(\sum_{}w_{kj}^{}f_{h}\left(\ldots \left(X^{p}\right)\right)\right)$ is the identified system of the target system.

So the goal is to find the appropriate value of the network weights **W**. This goal is achieved via learning which is performed by iteratively updating **W** such that the error **E**, at the output, is reduced. Different optimization techniques, like gradient-based back-propagation, genetic algorithm, and simulated annealing, are used for training neural networks. However, back-propagation [24] is the most commonly used algorithm for learning neural networks. It propagates the error backwards throughout the network layers and updates the weight by computing the gradient of the error. The derivative of the error with respect to the weights of the network is computed using the chain rule of differentiation.

The aim of our study is to model the protective effect of QCT against MTX-induced cytotoxicity in four different human cell lines (HeLa, PC-3, Hep G2, and SK-N-MC) using the above-mentioned multilayer neural network based system identification method. Toward this end, two ANN models are designed and trained to predict the protective effect of QCT against MTX-induced cytotoxicity.

## 3. Result

### 3.1. Protective Effect of QCT on MTX-Induced Cytotoxicity in HeLa, PC-3, HepG2, and SK-N-MC Cells

The experimental results revealed that % of inhibition increased with increasing dose of MTX, while QCT pretreatment significantly decreased % of inhibition in HeLa, PC-3, Hep G2, and SK-N-MC cell lines (Table 1). An increase in cell viability was observed in QCT-treated cells compared to MTX alone group (Figure 3). The result displayed that QCT at doses of 7.5 up to 20 μg/mL possessed the best protective effects. QCT alone treatment did not change the cell viability compared to the control group (Figure 3). QCT pretreatment also markedly decreased the MTX-caused LDH release (Figure 4). QCT alone treatment did not change the LDH activity compared to the control group (Figure 4).

### 3.2. Neuro Modeling of Protective Effect of QCT

The neuromodeling of protective effect of QCT against MTX (CTN, PAT, and ZEAR) in different cell lines (HeLa, PC-3, Hep G2, and SK-N-MC) has been performed via learning of neural networks. For this end, determination of % of inhibition and cell viability using crystal violet assay and LDH activity using LDH assay were performed.

#### 3.2.1. Neuro Modeling of Percentage (%) of Inhibition

The cytotoxicity was evaluated by determination of % of inhibition on HeLa, PC-3, Hep G2, and SK-N-MC cells that were pretreated with QCT (0–20 μg/mL) for 6 h, followed by the incubation with MTX (0–200 µM) for 24 h. The % of inhibition was measured for discrete amounts of MTX (0.0, 0.78125, 1.5625, 3.125, 6.25, 12.5, 25, 50, 100, and 200 µM) at every QCT dose of 0.0, 5.0, 7.5, 10, 15, and 20 µg/mL for each cell lines and MTX. A total 720 data points were measured, of which 480 data points corresponding to the % inhibition for QCT values of 0.0, 5.0, 10, and 20 µg/mL, as listed in Table 1 were used to train the neural network model whereas 240 data points corresponding to the % inhibition for QCT values of 7.5 and 15 µg/mL were reserved for testing. Out of the 480 data points, Table 1, used for training the neural network, 20% samples were randomly chosen and set aside for validation.

The doses of MTX and QCT are the primary inputs to the neural network, and the target to be learned is the corresponding % of inhibition. As the % inhibition of QCT on different cell lines and different MTXs is different, a naïve way would be to learn 12 different neural network models corresponding the four different cell lines and three different MTXs. However, in this study we aim to design a single neural network that models the overall behavior on the above mentioned cell lines and MTXs. To distinguish the % inhibition corresponding to the four cell lines and the three MTXs, additional binary codes are used as input to the network. The four cell lines HeLa, PC-3, Hep G2, and SK-N-MC are encoded as 1000, 0100, 0010, and 0001, respectively, and, the three MTXs (CTN, PAT, and ZEAR) are encoded as 100, 010, and 001, respectively. For example, the input to the neural network corresponding to 10 µg/mL QCT on HeLa cell lines for 0.78125 µM CTN is (1, 0, 0, 0, 1, 0, 0, 0.78125, 10). The doses of QCT and MTXs are normalized using their corresponding maximum doses used in this study, i.e., (dose of QCT/20, dose of MTX/200). The output of the network, i.e., the % of inhibition, is expressed a real number between zero and one [25].

So the neural network to be designed should have nine input and one output nodes. The appropriate number of hidden nodes required to learn the system was determined empirically. Starting with an initial five hidden nodes in a single hidden layer, the number of hidden nodes were incremented gradually. The network was trained using the back-propagation algorithm with a learning rate of 0.05 on the training dataset for a fixed 10,000 epochs, and its performance was evaluated on the randomly selected validation dataset. The root mean square error
(6)rmse=1N∑i=1N(yi−ti)2
was employed to evaluate the performance on the validation dataset where *N* is the number of validation data points. The results of the empirical method are presented in Table 2. The network with 20 and 30 hidden nodes produced the lowest error on the validation dataset. So, keeping the number of nodes in the first hidden layer at 20, another hidden layer was added to this network. Staring with three hidden nodes, the nodes in the second hidden layer were increased in steps of three. The network was then trained and validated as described earlier. The results of the empirical method for determining the number of nodes in the second hidden layer is presented in Table 3.

From Table 3, it is seen that the neural network with two hidden layers produces the lowest error in the validation dataset. Hence, a three-layered neural network of nine input nodes, 20 hidden nodes, 15 hidden nodes, and 1 output node, namely, a 9-20-15-1 network, as shown in Figure 5, was chosen as the optimal network to learn the % of inhibition for all the combinations of cell lines and MTXs. This empirically selected network was then finally trained on the whole training dataset (including the validation dataset). Figure 6 shows the error curve obtained during the learning of this network on the training and the test dataset, and the corresponding RMSEs are given in Table 4.

After the learning was completed, the network was evaluated on the test set and the % of inhibition values were obtained from the output of the ANN. The % of inhibition surface obtained from the trained neural network for Hep G2 cell lines with MTX CTN ranging from 0 to 200 µM for different doses of QCT ranging from 0 to 20 µg/mL is presented in Figure 7. Note that the increasing directions of QCT and CTN axis are indicated by arrow heads. The experimentally measured data is superimposed on this surface with solid circles. The black solid circles are the % of inhibition values used for training the ANN and the blue and red circles are those of test data. The solid circles hidden partially in the surface indicate that the % inhibition values predicted by the ANN are close to the experimentally measured data. For better visualization of the modeling accuracy of the ANN on the test data, the % inhibition values predicted by the ANN at QCT values of 7.5 and 15 µg/mL are presented in Figure 7c. The solid lines represent the outputs of ANN, whereas the solid dots are the measured data. Observe that the output of the ANN is close to the measured data at every test point.

The results of ANN were compared with that of linear regression method which is commonly used for a rough estimation of experimental results. Figure 7b shows the surface of the % of inhibition values obtained with linear regression of Hep G2 cell, and Figure 7d shows the % of inhibition curves at QCT values of 7.5 and 15 µg/mL concentrations.

Similar figures for other MTXs, such as PAT and ZEAR, are presented in Figure 8 and Figure 9, respectively. As seen in Figure 7, Figure 8 and Figure 9, the qualitative results of modeling the system with ANN shows excellent performance for the four different cell line types and three different MTXs consistently. A quantitative evaluation of the network’s performance on the test set exhibited high correlation (*R* = 0.999) with the experimentally measured data, which is substantially higher than the correlation obtained from other statistical method such as partial least squares (PLS) regression (*R* = 0.90). Also note that separate models of linear regression or the partial least squares regression have to be built for each of the cell line and MTX combination, whereas the proposed ANN models all the cell line MTX combinations in as single model. The plot of the error between the measured % of inhibition and the predicted values using the ANN model and PLS is as shown in Figure 10.

#### 3.2.2. Neuro-Modeling of Cell Viability and LDH Activity

Similar experiments, as those done for modeling the % of inhibition of QCT, were also conducted to model the cell viability and LDH activity. As in the case of % of inhibition, variation of cell viability and LDH activity was measured on cells treated with QCT (5, 7.5, 10, 15, and 20 μg/mL) and incubated with MTXs: CTN (100 µM), PAT (50 µM), and ZEAR (100 µM). As shown in Figure 3 and Figure 4, a total of 72 data points were measured for % of cell viability and LDH activity, respectively. Out of the total 72 data points, 48 data points were used to train the ANN, whereas the remaining 24 data points, corresponding to QCT values of 7.5 and 15 µg/mL, were reserved for testing. As before, 20% of samples chosen randomly from the training set were used for validating the neural network.

The aim of neural network in this case is to predict the cell viability and LDH activity given the doses of MTXs and QCT for all the combination of MTXs and cell lines. The architecture of the ANN for modeling this system is different from the case of % of inhibition model as the network needs to output two values. The output of the network, i.e., the cell viability and LDH activity, are normalized to real numbers between zero and one by dividing with their corresponding maximum values encountered in this study, i.e., 100 and 350, respectively. The binary encoding scheme as described in Section 3.2.1 is used to distinguish the different cell lines and the MTX combination. Hence the total number of input nodes is equal to 9. The optimal network architecture for this task was also determined by following the empirical method described in Section 3.2.1. The results of the empirical method are presented in Table 5 and Table 6.

A network with nine input nodes, 10 hidden nodes, 10 hidden nodes, and two output nodes, namely, a 9-10-10-2 network, shown in Figure 11, was determined to be the optimal network for this task. This network was then trained on the whole training dataset (including the validation dataset). Figure 12 shows the evolution of error during the learning of this network on the training and the test dataset, and the corresponding RMSEs are given in Table 7.

The trained network was evaluated on the test set and the cell viability and LDH activity values were obtained from the output of the ANN. The cell viability and the LDH activity curve obtained from the trained neural network for different cell lines pretreated with different doses of QCT ranging from 0 to 20 µg/mL and incubated with MTXs: CTN (100 µM), PAT (50 µM), and ZEAR (100 µM) is presented in Figure 13. The experimentally measured data is superimposed on this curve with markers, where ‘*’ indicates the data points used for training the ANN and ‘o’ for testing.

As seen from Figure 13, the qualitative results of modeling the system with ANN shows excellent performance for the 4 different cell line types and 3 different MTXs consistently. A quantitative evaluation of the network’s performance on the test set exhibited high correlation (*R*_cell_viability_ = 0.995 and *R*_LDH_activity_ = 0.997) with the experimentally measured data. 

The plot of the error between the measured cell viability and LDH activity and the corresponding values predicted using the ANN model is as shown in Figure 14.

## 4. Discussion

The main objective of this paper was to model the protective effect of QCT against MTXs using ANNs, and to verify the ability of the model in estimating the protective effect of QCT. Specifically, the protective effect of QCT against three different MTXs (Citrinin, Patulin, and Zearalenol) on four different cell lines (HeLa, PC-3, Hep G2, and SK-N-MC cell lines) was measured experimentally, and the data were used to model their nonlinear protective functions (% of inhibition, cell viability, and LDH activity) using multilayer neural networks.

The experimental measurements revealed that treatment with MTX significantly decreased cell viability and increased LDH activity. However, the % of inhibition of four different cells pretreated with the three MTXs was consistently decreased with the dose of QCT. Also, pretreatment with QCT attenuated MTX-induced alteration of cell viability and LDH activity that it could protect the cell lines from cytotoxicity. The effects of QCT against the MTX-induced cytotoxicity were conducted via cell viability and LDH release assays in Hela, PC-3, Hep G2, and SK-N-MC cell lines. The experimental results showed that treatment with MTX significantly decreased cell viability and increased LDH activity. However, pretreatment with QCT significantly attenuated MTX-induced alteration of cell viability and LDH activity, which suggests that it could protect the cell lines from cytotoxicity. Therefore, these results suggest that QCT may inhibit MTX-induced diseases in humans.

Two different ANN models with sizes of 9-20-15-1 and 9-10-10-2, determined empirically, were used to model the % of inhibition, the cell viability, and LDH activity, respectively. For both tasks, the experimentally measured data for the protective effects of QCT for three different MTXs in four different cells was used for training the neural network. As a result, twelve and twenty-four different models of precise protective effects of QCT were built on the two ANNs, respectively. Unlike the commonly used statistical methods, like linear regression or partial least square regression, which require separate models to be computed for each MTX and cell line combination, a single neural network was designed to model the different combinations using a special binary encoding scheme for each MTX and cell line combination. Moreover, quantitative evaluations of the network’s performance on the test sets exhibited high correlation with experimentally measured data which was substantially higher than that of individual models computed using other statistical methods.

It was observed that the additional burden for the neural network to discriminate between the different input combinations, expressed as binary codes, demands comparatively larger networks, which require more number of iterations for network convergence and hence longer training times. However, it was shown that single model for different input combination provides a unified and elegant solution with the ability to precisely model the protective effects of QCT against MTXs.

## 5. Material and Methods

### 5.1. Materials and Cell Culture

Citrinin (CTN), patulin (PAT), zearalenol (ZEAR), and quercetin (QCT) were purchased from Sigma-Aldrich (St. Louis, MO, USA). HeLa (cervical cancer cell), PC-3 (prostate cancer cell), Hep G2 (liver cancer cell), and SK-N-MC (brain cancer cell) human cell lines were obtained from the American type culture collection (Manassas, VA, USA). The cells were grown in Dulbecco’s Modified Eagle’s Medium (DMEM) and Roswell Park Memorial Institute (RPMI) 1640 supplemented with 10% fetal bovine serum (FBS), 4.5 g/L d-glucose, 2 mmol/L l-glutamine, 110 mg/L sodium pyruvate, 100 U/mL penicillin, and 100 μg/mL streptomycin at 37°C in a humidified atmosphere containing 95% air and 5% CO_2_.

To prevent mycoplasma effectively we followed a procedure described previously [26]. Briefly, we brought all cells and culture materials from reliable sources, used good aseptic technique, recommended antibiotic used for culture medium to eradicate all contamination and finally disinfecting the laminar flow hood after working.

### 5.2. Measurement of Mycotoxin-Induced Cytotoxicity

The MTX-induced cytotoxicity in different cell lines (HeLa, PC-3, Hep G2, and SK-N-MC) was measured by determination of percentage (%) of inhibition, cell viability, and LDH activity.

#### 5.2.1. Measurement of Percentage (%) of Inhibition

At first, we measured percentage (%) of inhibition to observe the cytotoxic effect of the MTXs (Citrinin, Patulin, and Zearalenol) and MTX + QCT on HeLa, PC-3, Hep G2, and SK-N-MC. Briefly, cells were seeded in 24-well plates with 5 × 10^4^ cells per well in culture media and allowed to attach overnight; cells were pretreated with QCT (0–20 μg/mL) at 37 °C in a humidified atmosphere of 5% CO_2_/95% air for 6 h followed by the incubation with mycotoxins (0–200 µM) for 24 h.

#### 5.2.2. Cell Viability

Crystal violet assay was used to determine MTX-induced cell death. Briefly, cells were seeded in 24-well plates with 5 × 10^4^ cells per well in culture media and allowed to attach overnight. The cells were pretreated with the doses of QCT at 5, 10, and 20 μg/mL at 37 °C in a humidified atmosphere of 5% CO_2_/95% air for 6 h, followed by the incubation with CTN (100 µM), PAT (50 µM), and ZEAR (100 µM) for 24 h. After 24 h of incubation, removed medium and washed the cells with phosphate buffer solution (PBS) and 0.2% crystal violet solution was added to each well. After 10 min of incubation, the crystal violet solution was removed carefully by washing with water. Finally, added 100 μL 1% sodium dodecyl sulfate (SDS) to solubilize the color solution until the color is uniform and no areas of dense coloration in bottom of wells. The samples were read at 590 nm in a microplate reader (Spectra MAX, Gemini EM, Molecular Device, Sunnyvale, CA, USA). The cell viability is expressed as the percentage of absorbance of control.

#### 5.2.3. Lactate Dehydrogenase (LDH) Activity

Lactate dehydrogenase (LDH) activity assay was used to determine MTX-induced cytotoxicity. LDH release into the media was taken as an indicator of cell damage and the assay is based on the principle of reduction of nicotinamide adenine dinucleotide (NAD) by LDH. The reduced NAD (NADH) is utilized in the stoichiometric conversion of a tetrazolium dye which is measured spectrophotometrically using an LDH assay kit (Cat. No. 04744926001, Sigma, Saint Louis, MO, USA). Briefly, cells were seeded (5 × 10^4^ cells/well) and cultured in 24-well culture plates. The cells were then preincubated with or without different concentrations of QCT (5, 7.5, 10, 15, and 20 μg/mL) at 37 °C for 6 h followed by incubation with CTN (100 µM), PAT (50 µM), and ZEAR (100 µM) for 24 h. After treatment was over, cells were centrifuged at 240 × *g* for 4 min and the culture supernatant was transferred in a new plate. The assay mixture was prepared and added to each well and the plate incubated wrapped in foil at room temperature for 30 min. Reaction was terminated by adding the stop solution to each well. The plate was read at 490 nm at a reference wavelength of 690 nm. The extent of cytotoxicity is expressed as the percentage of absorbance of control.

### 5.3. Statistical Data Analysis and Neural Network Training

All the data were expressed as mean ± SD and one-way ANOVA (Analysis of variance) followed by Dunnett’s test was used for the statistical analysis using SPSS software (version 16, SPSS, Inc., Chicago, IL, USA). * *p* < 0.05 and ** *p* < 0.01 were considered significant. The artificial neural network was trained using custom codes developed by the authors written in MATLAB(R) (2017b, Mathworks, Natick, MA, USA) on a standard computer station (Intel(R) Core(TM) i7-6700k 4.00 GHz, 8 cores, 8 GB RAM) machine, whereas the partial least squares method was implemented using MATLAB’s built-in *plsregress* function.

## Figures and Tables

**Figure 1 ijms-20-01725-f001:**
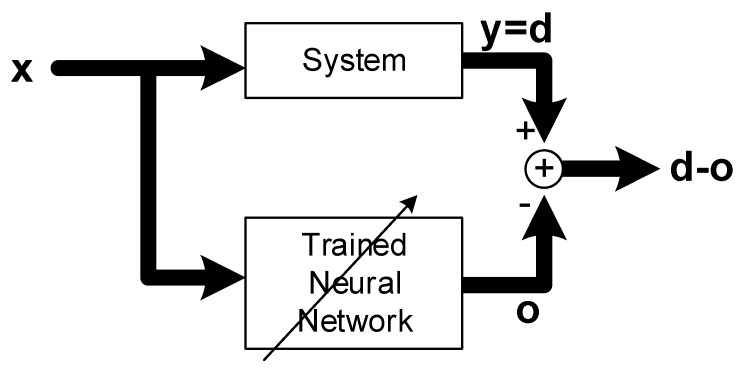
Concept of system identification with artificial neural networks, where lines with arrow heads denote parameter adjustment via learning.

**Figure 2 ijms-20-01725-f002:**
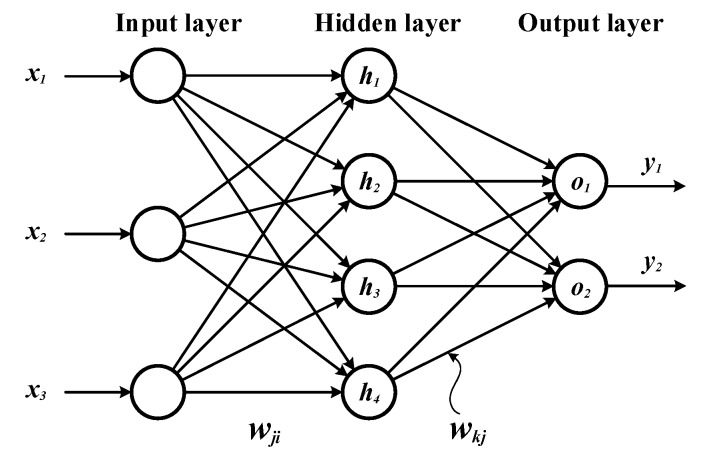
Architecture of our artificial neural network.

**Figure 3 ijms-20-01725-f003:**
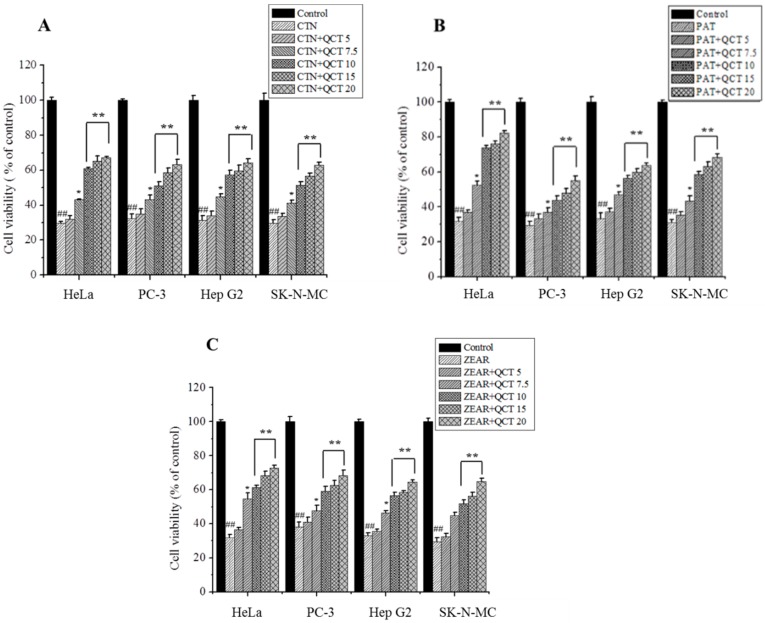
The protective effect of QCT against MTX-induced cell death. (**A**) Cell viability on HeLa, PC-3, Hep G2, and SK-N-MC cells when pretreated with QCT and post-treated with various MTXs, such as (**A**) CTN, (**B**) PAT, and (**C**) ZEAR. Values were represented as mean ± SD. ^##^
*p* < 0.01 as compared with the control group; * *p* < 0.05; ** *p* < 0.01 as compared with the MTX alone group.

**Figure 4 ijms-20-01725-f004:**
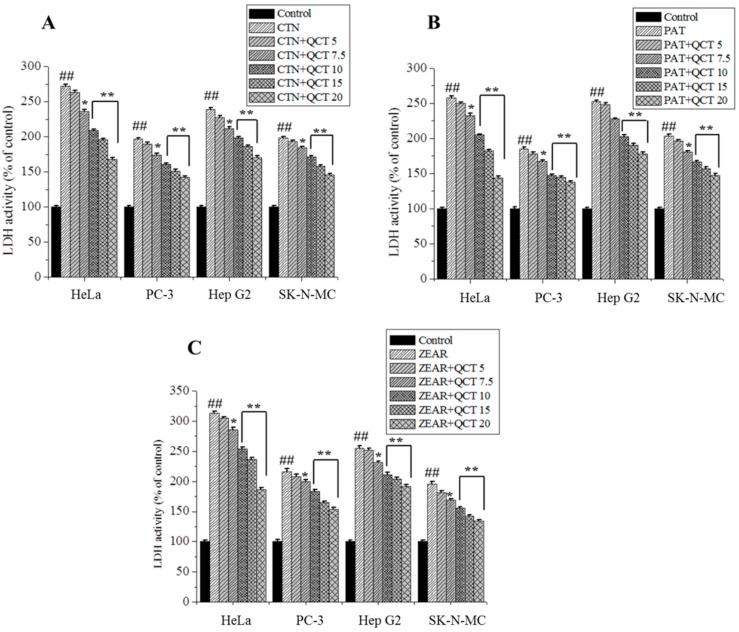
The protective effect of QCT against MTX-induced cytotoxicity. LDH activity on HeLa, PC-3, Hep G2, and SK-N-MC cells when pretreated with QCT and post-treated with various MTXs, such as (**A**) CTN, (**B**) PAT, and (**C**) ZEAR. Values were represented as mean ± SD. ^##^
*p* < 0.01 as compared with the control group; * *p* < 0.05; ** *p* < 0.01 as compared with the MTX alone group.

**Figure 5 ijms-20-01725-f005:**
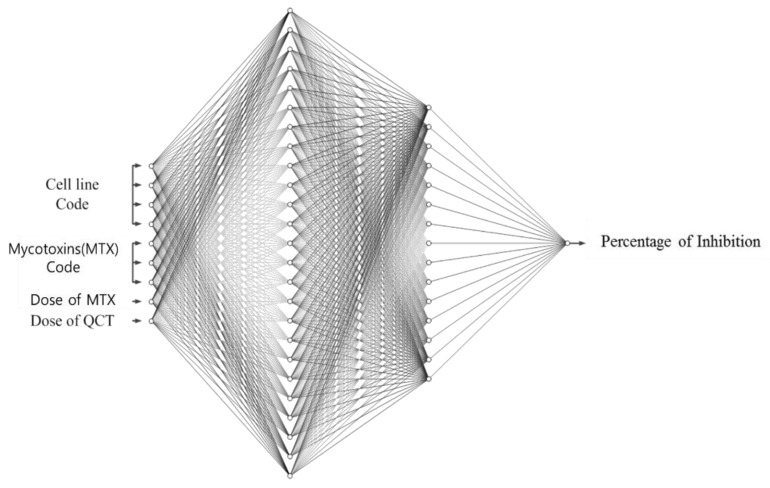
Architecture of artificial neural network (ANN) (9-20-15-1) for learning of % of inhibition. Biases of nodes are omitted in this figure, where four input terminals are used for indicating cell lines and three input terminals are to determine MTXs and other two input terminals are for the dose of MTX and QCT, respectively. Also, the single output terminal is for % of inhibition.

**Figure 6 ijms-20-01725-f006:**
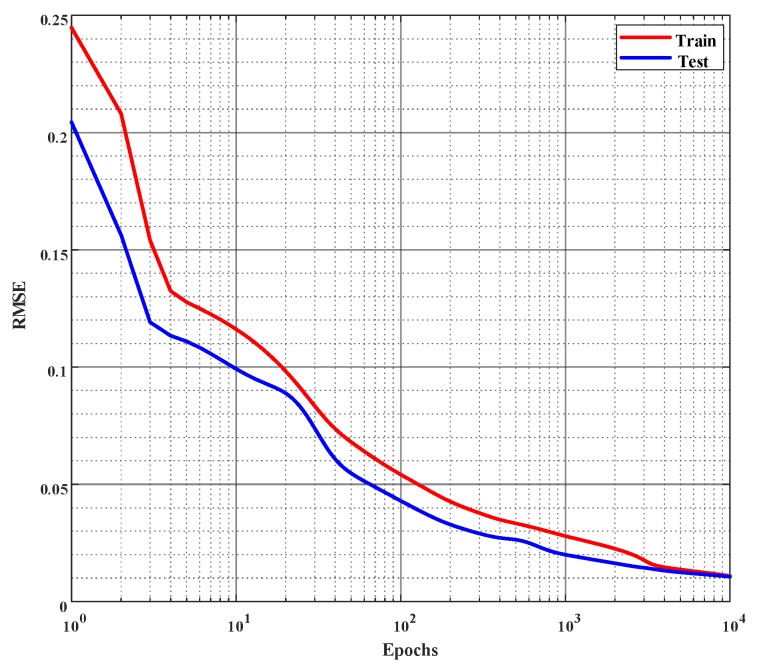
Error curves of learning % of inhibition on the training and the test set. Root mean square error (*rmse*) is employed to evaluate the performance and the network is trained for 10,000 epochs or until the *rmse* reaches 0.01, whichever occurs first. (The *x*-axis is plotted in log scale.)

**Figure 7 ijms-20-01725-f007:**
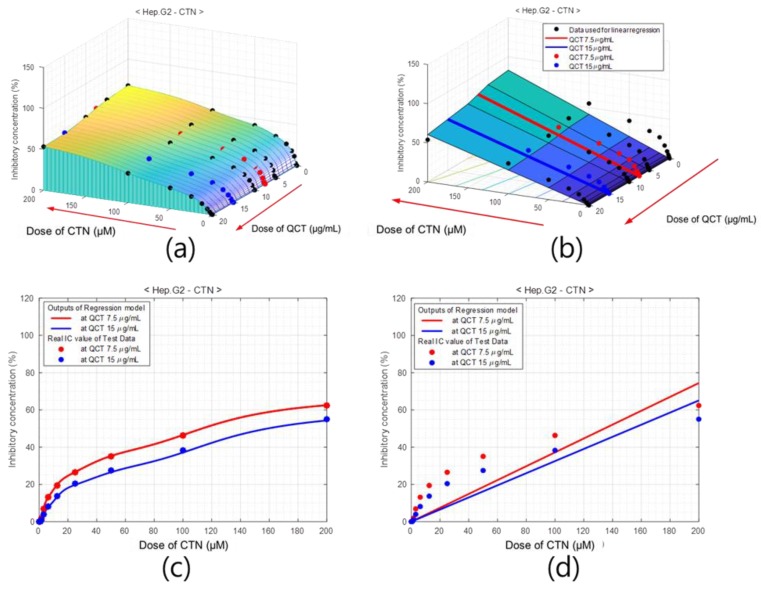
Percentage of inhibition model of Hep G2 cell for dose of QCT and CTN. (**a**) Surface of % of inhibition values obtained with ANNs, where black dots indicate training data and red and blue dots are test data. (**b**) % of inhibition surface obtained with linear regression. Cross-sections of (**c**) ANN-based model and (**d**) the linear regression model at QCT = 7.5 μg/mL (red) and QCT = 15 μg/mL (blue), which comprise the test area. The outputs of the neuro-model are close to the experimentally measured % of inhibition values.

**Figure 8 ijms-20-01725-f008:**
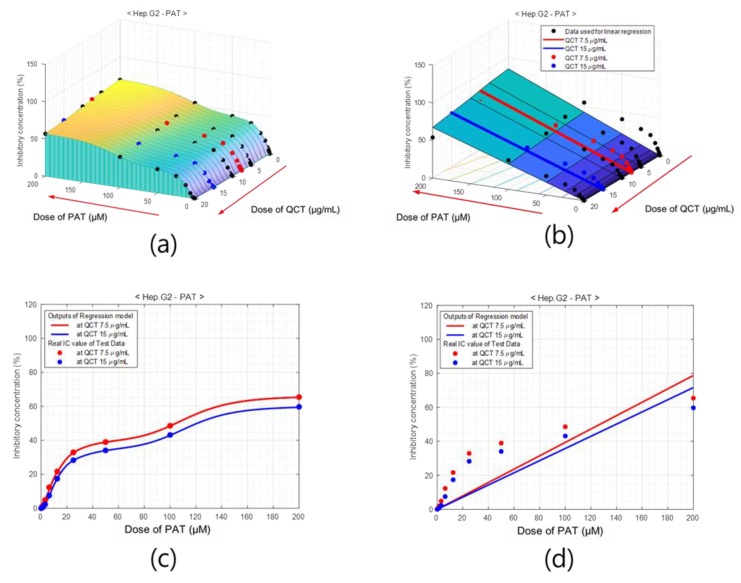
Percentage of inhibition model of QCT for Hep G2 cell with PAT. Surface of % of inhibition obtained using (**a**) ANN model and (**b**) linear regression. Cross-sections of (**c**) ANN-based model and (**d**) linear regression model at QCT = 7.5 μg/mL (red) and QCT = 15 μg/mL (blue), which comprise the test area.

**Figure 9 ijms-20-01725-f009:**
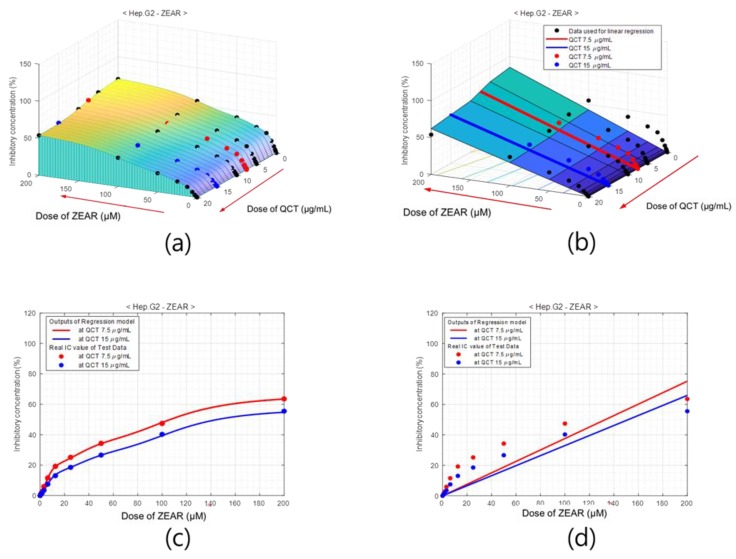
Percentage of inhibition model of QCT for Hep G2 cell with ZEAR. Surface of % of inhibition obtained using (**a**) ANN model and (**b**) linear regression. Cross-sections of (**c**) ANN-based model and (**d**) linear regression model at QCT = 7.5 μg/mL (red) and QCT = 15 μg/mL (blue), which comprise the test area.

**Figure 10 ijms-20-01725-f010:**
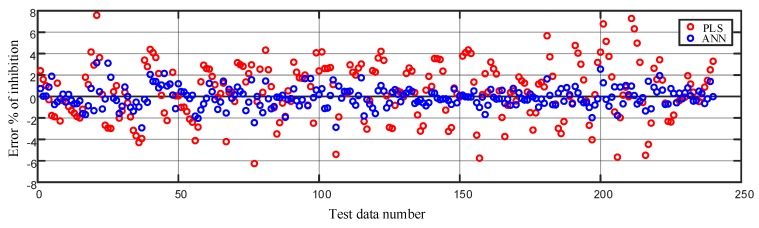
Comparison of error in modeling the % of inhibition on the test data using the partial least squares regression (PLS) and the artificial neural network method.

**Figure 11 ijms-20-01725-f011:**
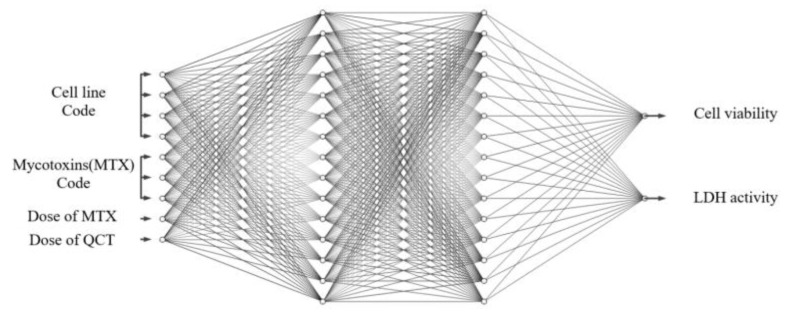
Architecture of ANN which was used for the modeling of cell viability and LDH activity. The size of the ANN is 9-10-10-2. Biases of nodes are omitted in this figure.

**Figure 12 ijms-20-01725-f012:**
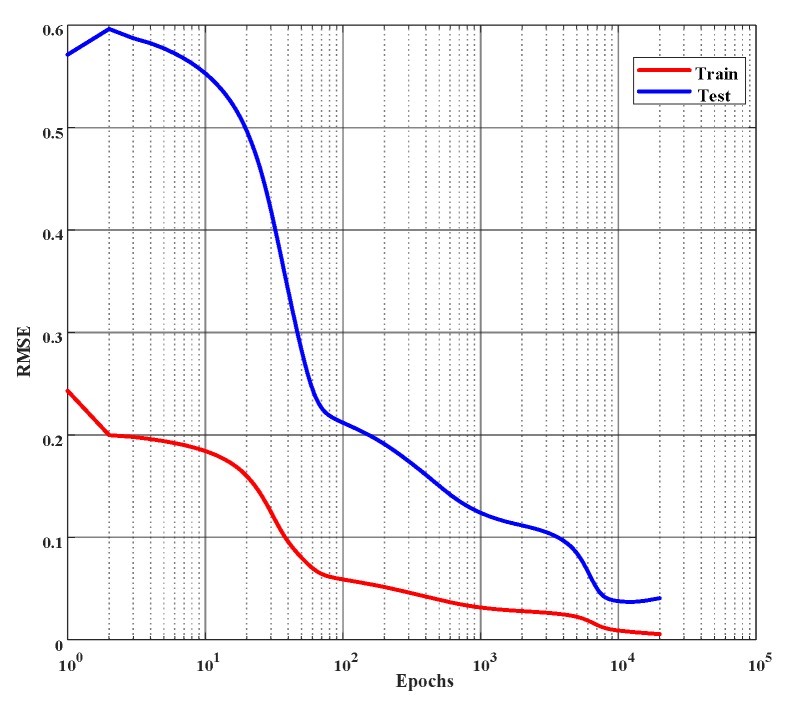
Error curves for the learning of cell viability and LDH activity on the training and the test set. Root mean square error (*rmse*) is employed to evaluate the performance and the network is trained for 20,000 epochs or until the *rmse* reaches 0.001, whichever occurs first. (*x*-axis is plotted in log scale.)

**Figure 13 ijms-20-01725-f013:**
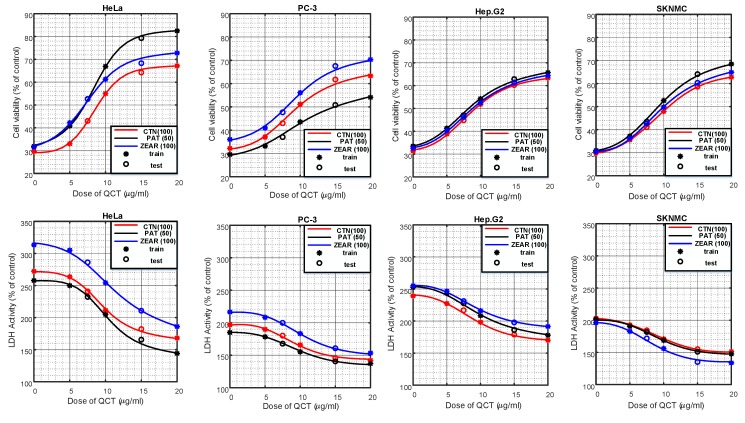
Cell viability and LDH activity predicted by the neuro-model on different cell lines pretreated with QCT and post-treated with various MTX: CTN (100 µM), PAT (50 µM), and ZEAR (100 µM). ‘*’ marks the locations of training data whereas ‘o’ marks the location of test data.

**Figure 14 ijms-20-01725-f014:**
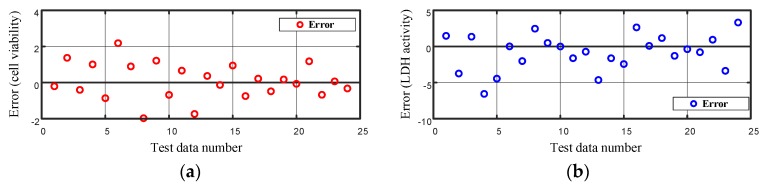
Error in modeling the (**a**) cell viability and (**b**) LDH activity on the test data using the artificial neural network.

**Table 1 ijms-20-01725-t001:** Measured % of inhibition for neural network learning.

Measured % of Inhibition													
Types of Cell	Types of MTX		Dose of MTX (µM)	0	0.78125	1.5625	3.125	6.25	12.5	25	50	100	200
		Dose of QCT (µg/mL)											
HeLa [25]	CTN	0.0		0.000	4.037	7.148	12.442	18.547	25.711	34.773	62.960	73.363	77.878
		5.0		0.000	2.552	4.725	8.745	14.436	21.066	29.509	55.722	65.976	72.426
		10.0		0.000	1.207	2.252	4.037	7.148	11.442	17.136	34.121	48.016	60.865
		20.0		0.000	0.663	1.207	2.157	3.601	5.615	8.251	19.558	33.847	45.154
	PAT	0.0		0.000	10.394	20.702	30.119	39.728	48.266	56.931	66.120	86.211	94.497
		5.0		0.000	8.528	16.427	24.212	31.912	41.947	48.993	58.207	79.298	89.532
		10.0		0.000	3.379	5.742	9.746	12.900	19.961	26.679	33.080	58.963	73.549
		20.0		0.000	0.630	1.805	4.154	5.931	11.134	16.346	21.520	42.283	60.139
	ZEAR	0.0		0.000	5.744	11.029	17.033	25.506	32.597	38.322	47.308	68.444	92.675
		5.0		0.000	4.753	8.253	14.074	20.551	27.415	31.961	39.877	57.776	80.446
		10.0		0.000	0.467	2.256	3.928	7.271	13.851	18.196	22.502	38.721	62.941
		20.0		0.000	0.396	0.829	2.104	3.426	7.433	10.569	14.425	27.220	52.217
PC-3	CTN	0.0		0.000	2.746	5.031	8.530	13.621	20.016	35.176	44.174	53.834	68.469
		5.0		0.000	2.127	4.169	6.931	10.859	17.299	33.881	42.561	51.789	66.803
		10.0		0.000	1.175	2.391	5.324	7.588	12.443	31.828	38.841	48.858	63.137
		20.0		0.000	0.595	1.422	3.591	5.422	8.888	26.740	34.864	43.502	57.151
	PAT	0.0		0.000	3.871	7.305	15.559	23.058	30.820	41.563	57.171	70.441	83.875
		5.0		0.000	3.597	5.861	11.980	19.284	25.785	37.683	55.698	66.764	80.874
		10.0		0.000	2.556	3.847	6.964	11.164	19.272	29.085	49.379	58.440	73.241
		20.0		0.000	1.120	2.094	3.409	5.186	12.736	22.571	39.810	48.884	66.569
	ZEAR	0.0		0.000	3.348	4.754	7.981	16.767	31.562	40.523	51.779	60.057	69.167
		5.0		0.000	3.044	3.999	6.097	13.233	28.497	37.270	47.886	56.613	65.615
		10.0		0.000	2.188	2.750	3.923	8.593	21.893	30.612	40.493	49.285	57.291
		20.0		0.000	1.802	1.921	2.129	3.302	11.109	20.825	31.898	40.543	48.546
Hep G2	CTN	0.0		0.000	0.357	2.480	11.010	17.209	24.668	32.108	41.636	52.204	67.657
		5.0		0.000	0.278	2.293	8.824	14.740	21.952	29.435	38.582	49.717	65.390
		10.0		0.000	0.135	1.160	5.596	11.123	17.181	23.743	31.516	42.822	59.160
		20.0		0.000	0.022	0.777	2.847	6.236	11.694	18.509	25.229	35.645	53.168
	PAT	0.0		0.000	2.571	5.199	8.137	16.763	25.955	37.198	43.070	53.062	69.132
		5.0		0.000	1.695	3.327	5.707	11.952	23.024	34.639	40.480	50.132	67.178
		10.0		0.000	0.879	1.535	3.295	10.396	19.582	31.226	36.960	46.394	63.472
		20.0		0.000	0.250	0.565	1.304	6.122	16.175	26.296	31.880	40.779	56.756
	ZEAR	0.0		0.000	2.451	4.555	8.583	15.487	25.169	31.310	40.590	55.180	69.852
		5.0		0.000	1.851	3.565	6.963	13.048	22.278	28.300	37.765	51.332	66.941
		10.0		0.000	1.291	2.495	4.820	9.110	16.094	21.458	30.619	44.090	59.257
		20.0		0.000	0.682	1.471	3.153	6.551	12.379	17.782	25.190	38.595	53.895
SK-N-MC	CTN	0.0		0.000	5.149	9.054	15.030	24.020	33.060	44.460	54.860	70.190	89.515
		5.0		0.000	4.116	7.303	12.658	20.801	29.273	41.660	52.521	67.797	86.988
		10.0		0.000	2.554	4.578	8.514	14.904	23.242	35.788	44.936	59.906	77.792
		20.0		0.000	1.256	2.702	5.280	9.363	16.641	27.459	35.506	49.598	63.050
	PAT	0.0		0.000	3.830	7.512	13.157	22.759	39.909	52.706	69.069	78.993	89.176
		5.0		0.000	3.215	6.264	11.385	20.793	36.114	49.039	64.793	74.542	85.105
		10.0		0.000	2.680	4.963	9.308	17.370	31.163	44.130	59.581	68.837	79.193
		20.0		0.000	1.585	3.099	6.162	12.077	21.915	31.908	48.888	58.464	64.169
	ZEAR	0.0		0.000	6.051	10.256	14.708	19.469	26.018	36.300	53.844	75.113	80.588
		5.0		0.000	4.359	7.981	11.581	15.744	22.170	32.511	49.840	72.063	78.304
		10.0		0.000	2.279	4.224	6.678	10.003	15.094	24.568	40.715	65.506	72.174
		20.0		0.000	1.370	2.463	4.262	6.809	10.899	18.943	33.655	55.343	60.131

MTX: Mycotoxin, QCT: Quercetin, CTN: Citrinin, PAT: Patulin, ZEAR: Zearalenol.

**Table 2 ijms-20-01725-t002:** Performance comparison of a single hidden layer neural network by varying the number of nodes in the hidden layer.

No. of Hidden Nodes	Train RMSE	Validation RMSE
5	0.0302	0.0282
7	0.0247	0.0278
10	0.0223	0.0245
12	0.0202	0.0208
15	0.0214	0.0198
17	0.0200	0.0211
**20**	0.0197	**0.0190**
22	0.0203	0.0194
25	0.0200	0.0236
27	**0.0187**	0.0192
30	0.0188	**0.0190**
32	0.0193	0.0203

**Table 3 ijms-20-01725-t003:** Performance comparison of a two hidden layer neural network by varying the number of nodes in the last hidden layer. The first and the second elements of the No. of Hidden Nodes are node numbers of the first and the second hidden layers, respectively.

No. of Hidden Nodes	Train RMSE	Validation RMSE
(20, 3)	0.0129	0.0150
(20, 6)	0.0110	0.0173
(20, 9)	0.0109	0.0162
(20, 12)	0.0113	0.0161
**(20, 15)**	0.0106	**0.0147**
(20, 18)	0.0105	0.0153
(20, 21)	0.0108	0.0151

**Table 4 ijms-20-01725-t004:** Performance of the final 9-20-15-1 network on the train and test dataset.

Network	Train RMSE	Test RMSE
9-20-15-1	0.0110	0.0111

**Table 5 ijms-20-01725-t005:** Performance comparison of a single hidden layer neural network by varying the number of nodes in the hidden layer.

No. of Hidden Nodes	Train RMSE	Validation RMSE
5	0.0253	0.0597
7	0.0226	0.0943
**10**	0.0174	**0.0540**
12	0.0175	0.0737
15	0.0192	0.0721
17	0.0197	0.0937
20	0.0100	0.0545
22	0.0190	0.0830
25	0.0175	0.0892
27	0.0205	0.0852
30	0.0116	0.0637
32	0.0193	0.0851

**Table 6 ijms-20-01725-t006:** Performance comparison of a two hidden layer neural network by varying the number of nodes in the last hidden layer. The first and the second elements of the No. of Hidden Nodes are node number of the first and the second hidden layers, respectively.

No. of Hidden Nodes	Train RMSE	Validation RMSE
(10, 2)	0.0100	0.0612
(10,4)	0.0100	0.0450
(10, 6)	0.0100	0.0384
(10, 8)	0.0114	0.0338
**(10, 10)**	0.0100	**0.0337**

**Table 7 ijms-20-01725-t007:** Performance of the final 9-10-10-2 network on the train and test dataset.

Network	Train RMSE	Test RMSE
9-10-10-2	0.0052	0.0364

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
