# Peer review of "Precise Modeling of the Protective Effects of Quercetin against Mycotoxin via System Identification with Neural Networks"

_ijms, 2019, doi:10.3390/ijms20071725_

Round 1
Reviewer 1 Report
The authors has addressed all issues.
Author Response
Thank you very much for your review.
We checked and amended English language in green.

Reviewer 2 Report
The revised version of the paper contains additional information, which is important for a better understanding of the experiments and results. I have these comments which will hopefully contribute to improve the paper.
1) It seems the authors set aside one third of the dataset for testing and additionally 20 % for validation. Considering the size of the networks, the training set is perhaps too small, compared to the validation and test sets, and to what is more common in the literature (30-20% for test and validation). Maybe the authors could justify the decision.
2) The paper does not specify the type of neural networks, transfer functions or learning methods used for training. This is specially important considering the networks seem to be very large and learn very slowly.
3) The learning rate 0.05 may be too high for the type of data. Even though the evolution of the errors (Figures 6, 12) seems "typical", it is surprising that for just a few hundred or tens of samples it takes 10,000 or 20,000 epochs of training. So I wonder if the learning rate is adequate. Have the authors tried other learning rates?
Is the error shown in Figure 6/12 the test set? Or which set is it?
4) As written above, the training time and architecture of the networks seem very large for the type and size of the dataset. They may be correct, of course. But for a complete analysis, and to substantiate the quality of the model, the authors should also present:
-- The evolution of the errors during training for the three sets (train, validation and test), not just for one of them;
-- The RMSE for all the three sets (train, validation and test) in Tables 2, 3, 4, 5.
This additional information will show there is no overfitting of the data, give a better insight on the adequacy of the learning rate, and rule out problems like vanishing or exploding gradient during learning.
5) A difference of 5 neurons between models may be a big leap for the big network model. The authors should justify if there is a reason for such a big leap, specially considering the aim of the paper is to create a "precise model".
6) Minor issues:
-- There are some typos, like "the dose[s] of QCT and MTX are..."
-- The authors might also mention methods of optimizing the network structures, like genetic algorithms.
Author Response
We thank the reviewer for his constructive comments. We have revised the paper by incorporating the suggested changes. We believe that the reviewer’s comments have helped us to improve the quality of the manuscript.
We upload our response as a word file for your comments.

Round 2
Reviewer 2 Report
The paper is more complete now.
Some minor issues:
-- I still think the number of neurons not being tested in steps of one or two neurons each time, is a major weakness of the paper.
-- MSE for the test sets should be in the tables, although I also understand the Figures show the evolution of performance in the test set.
-- I still found one typo, "simulated annealing for used for training neural networks".
A major issue:
I understand all layers of the network use sigmoid transfer functions. In general that should be avoided and that is probably the reason why such a high learning rate and so many epochs of training are required. That may also explain the weird shape of the learning performance.
The model is most probably suffering from the vanishing gradient problem. The sigmoid function in multi-layer networks is notoriously prone to cause the vanishing gradient, since its output is only in a narrow interval regardless of the change in the inputs. So the gradient of the error vanishes and learning is very slow, if not totally impaired.
I suggest the authors try using for example a linear transfer function (relu) in the first or last layers and maintain the sigmoid only for one of the layers. That should allow for much faster training, even with a lower learning rate, and a much smoother learning curve. This means repeating the experiments, but that should be fast in Matlab and the model should be much simpler and much faster in the end. I hope this helps.
Author Response
Dear Reviewer,
Thank you for your valuable comments.
Please find the attached file for review.
